# Input–Output Efficiency of the Beijing–Tianjin Sandstorm Source Control Project and Influencing Factors

**Yuxin Cui, Xuesong Gu \*, Zelin Liu and Jingxiong Yi**

School of Economics and Management, Beijing Forestry University, Beijing 100083, China; cuiyuxin20@bjfu.edu.cn (Y.C.); liuzelin@buaa.edu.cn (Z.L.); yijingxiong@stu.pku.edu.cn (J.Y.)
\* Correspondence: guxuesong@bjfu.edu.cn

**Abstract:** From the perspective of economic benefit, this paper uses the DEA method to measure the input–output efficiency of the Beijing–Tianjin sandstorm source control project in Beijing, Hebei, Shanxi, and Inner Mongolia. The results show that from 2003 to 2019, the efficiency of the four regions revealed a U-shaped trend, first decreasing and then increasing, which reflects the lagging characteristics of forestry engineering. The TFP of the sand source control project in these four places was calculated; the average efficiency growth rate was 1.4%, and it was found to be affected greatly by the rate of technological progress. The Tobit model was used to explore the influencing factors, and the results show that educational input and economic development level have a great effect on efficiency, fiscal pressure has a small effect on efficiency, and financial development level has a restraining effect on efficiency.

**Keywords:** sand source control project; input–output efficiency; influencing factors; DEA–Tobit model

## 1. Introduction

At the end of last century, land desertification in north China was serious, and sandstorms began to invade Beijing and Tianjin frequently, with a serious impact on the ecological environment of the region. In order to solve the problem of frequent sandstorm weather in Beijing, Tianjin, and even North China, the State Council approved the implementation of the Beijing–Tianjin sandstorm source control project. At present, the Beijing–Tianjin sandstorm source control project is in the process of implementing the second phase. The first phase of the project was launched in 2000 and lasted for 12 years, covering 75 counties and cities in five provinces, municipalities, and autonomous regions, including Beijing, Tianjin, Hebei, Shanxi, and Inner Mongolia. According to the Beijing–Tianjin Sandstorm Source Control Project Plan (2000–2010), the project aims to prevent further deterioration of the ecological environment through the protection of existing vegetation, sand blocking and afforestation, afforestation by air seeding, returning farmland to forest, grassland management, and comprehensive management of small watersheds. By 2012, a total of RMB 47.9 billion had been invested in the first phase of the Beijing–Tianjin sandstorm source control project. In the same year, the State Council announced the second phase of the Beijing–Tianjin Sandstorm Source project, which will be extended to 138 counties and cities in six provinces with a total investment of RMB 87.792 billion. The second phase of the project pays more attention to the comprehensive construction level of the project area, that is, on the basis of ensuring the expansion of afforestation area, the second phase of the project strengthens the construction of a series of supporting facilities to consolidate the achievements of the first phase. The second phase of the project promotes and unifies ecological, social, and economic benefit, combining the goal of ecological construction with that of improving local economic conditions. There are many specific initiatives, such as helping farmers and herdsmen who previously relied on indiscriminate cultivation and grazing for their livelihoods to change their production and lifestyle through fiscal subsidies, optimising the industrial structure while lifting farmers and herdsmen out of

poverty, and promoting the healthy development of agriculture and animal husbandry in the region.

In recent years, the policy goal of forestry key engineering projects has no longer been just to protect the ecological environment, but to emphasize more and more economic benefits in order to achieve a win–win situation between ecological benefits and social and economic benefits. Wang et al. conducted an empirical analysis of the construction effect of the Beijing–Tianjin sandstorm source control project and believed that both its ecological value and its local economic benefits should be considered in the construction process [1]. Zhou believed that the implementation of the Beijing–Tianjin sandstorm source control project promoted the adjustment of industrial structure in northwest Shanxi Province, and farmers' income significantly increased [2]. Zhang found that the implementation of ecological engineering promoted regional economic development; the Engel coefficient of herdsmen and peasant families in Inner Mongolia decreased significantly, and the production capacity of animal husbandry increased significantly as well [3]. Obviously, as a key forestry project, the Beijing–Tianjin sandstorm source control project is no longer only the original pure ecological project, and its economic benefits are increasingly prominent. In fact, it is very important to attach importance to the economic benefits of forestry engineering in order to consolidate the existing construction achievements and realize the sustainable development of forestry engineering. Yang et al. found that the sustainable development of forestry ecological engineering construction must be based on the "triple bottom line", that is, the coordinated development of economic wealth, social well-being, and ecological balance [4]. Xu Haili evaluated the comprehensive benefits of the Beijing–Tianjin sandstorm source control project in Shanxi Province from four aspects: forestry, agriculture, water conservancy, and ecological migration. After analysis, she learned that the project could significantly improve the local economic development conditions and environmental pollution in the actual control process [5]. To date, most studies have only focused on the ecological benefits of the Beijing–Tianjin sandstorm source control project, and there are few empirical studies on the input–output efficiency of the project in terms of economic benefits.

In the existing studies on the measurement of forestry input–output efficiency, domestic and foreign scholars alike have mostly used the DEA method to measure efficiency (Mlynarski et al. [6]; Cheng et al. [7]; Chiang Kao [8]; Yang et al. [9]). The DEA method mainly carries out evaluation through a data and mathematical programming model, which avoids the subjective consciousness of the evaluators to an extent in order to obtain more objective conclusions. In addition, the DEA method does not take dimension into account, which makes it convenient to use and simplifies the process of evaluation. The DEA method has relatively clear economic significance in that it can reflect the effectiveness of the production activities of the evaluated unit and further decompose this effectiveness into scale effectiveness and technical effectiveness so as to comprehensively evaluate the production status of the decision-making unit. In addition to DEA, Salehirad et al. studied the investment efficiency of forestry funds by using the analysis method of an input–output table [10]; however, this method mainly shows the input and output of each department of a system, the source of input, and the direction of output, as well as its technical and economic relationship. It cannot intuitively display the input–output efficiency and compare different systems. Ying used a vector autoregressive model to analyze the relationship between forestry fiscal expenditure and total forestry output value [11]. Deng et al. used a comparative analysis of indicators to analyze the differences in input–output characteristics and input–output efficiency of forestry in China and the United States [12]. However, this method was applied to the prediction of a correlated time series system and the study of the dynamic influence of random disturbance on a system of variables. It is not applicable to the case in this paper, where efficiency is calculated according to multiple input and output variables. In terms of index selection, Liu et al. stated that the Beijing–Tianjin sandstorm source control project had a significant positive effect on labor force transformation from the perspective of labor force structure [13]. Tian et al. believed that forestry labor input

and the output value of forestry primary industry had a great impact on the efficiency of forestry input and output in China [14]. Lai et al. calculated the input–output efficiency of forestry in Guangdong Province by using input indicators such as forestry land area and the forestry budget of the government and output indicators such as gross forestry output value and area of forest reconstruction [15]. As for the influencing factors, Yu et al. analyzed the main factors affecting efficiency through a Tobit regression model and concluded that tax sharing reform has a negative impact on fiscal expenditure efficiency, while forest tenure system reform has a positive impact on forestry fiscal expenditure efficiency [16]. Cao et al. reached a similar conclusion using a Tobit regression model, stating that both fiscal support and forest tenure reform had a positive impact on forestry production efficiency, while forest tenure reform did not positively regulate the relationship between fiscal support and forestry production efficiency [17]. Zadmirzaei et al. used a DEA and ND model to measure the relative performance of 24 Iranian forest management units and analyzed the impact of external involuntary factors on the technical efficiency of these units [18]. Yang et al. believed that due to the lack of capital formation mechanism in rural areas, the improvement of financial efficiency actually had a negative impact on rural capital efficiency [19]. Robson et al. believed that institutional and managerial factors are important factors affecting the efficiency of forestry industry [20]. Feng et al. concluded that labor force and capital have a significant positive impact on the growth of forestry industry, and as such it is necessary to increase the labor force and capital investment [21]. Li believes that forestry investment is an important factor affecting forestry output efficiency [22].

On the one hand, the current empirical research on the economic benefits of forestry projects has mostly been carried out from the perspective of evaluating the overall fiscal expenditure of forestry, and the research on the input–output efficiency of each forestry ecological governance project in terms of their respective economic benefits is insufficient. This paper studies the input–output efficiency of the economic benefits of the Beijing–Tianjin sandstorm source control project and analyzes and studies the effect of the Beijing–Tianjin sandstorm source control project on the local economy while achieving ecological protection. On the other hand, the existing studies on the efficiency of the Beijing–Tianjin sandstorm source control project mostly focus on its ecological benefits, while there are few empirical studies on its economic benefits. Moreover, the data used in the existing literature mostly date to before 2012, which cannot well reflect the results of the second phase of the project after 2012. In contrast, the data used in this paper have a much larger time frame, and better reflect the results of the Beijing–Tianjin sandstorm source control project.

At the end of the second phase of the Beijing–Tianjin sandstorm source control project, has the project achieved ecological protection and promoted local economic development? What are the factors affecting the input–output efficiency of the Beijing–Tianjin sandstorm source project? In order to solve these problems, this paper takes the afforestation area of barren mountain (sand) land, afforestation area of closed mountain (sand) land at the end of the year, national forestry investment, and other forestry investment as input indices, and takes the net income of farmers and the proportion of local primary industry as output indices. Based on the panel data of four provinces (municipalities and autonomous regions) from 2003 to 2019, the input–output efficiency of the Beijing–Tianjin sandstorm source project was calculated from both static and dynamic perspectives. The four variables of fiscal pressure, input in education, regional economic conditions, and financial development in the four provinces of Beijing, Hebei, Shanxi, and Inner Mongolia from 2003–2019 were used as explanatory variables, and the input–output efficiency values of the Beijing–Tianjin Sandstorm Source Control Project in the four provinces (municipalities and autonomous regions) were used as explanatory variables for regression analysis using a Tobit panel analysis with fixed effects. The regression results were used to find ways to optimize the efficiency.

In the Section 2, we propose the DEA method for calculating input–output efficiency, construct the Tobit regression model for analyzing the influencing factors, and explain the selected variables and data sources.

In the Section 3, we conduct an empirical analysis of input–output efficiency, including the calculation of static efficiency, the calculation of the change rate of economic efficiency, and the analysis of decomposition efficiency values.

The Section 4 conducts an empirical study of the influencing factors.

The Section 5 makes relevant policy recommendations based on the empirical findings.

The Section 6 summarises the full text and draws conclusions.

The algorithmic structure is as follows (Figure 1).

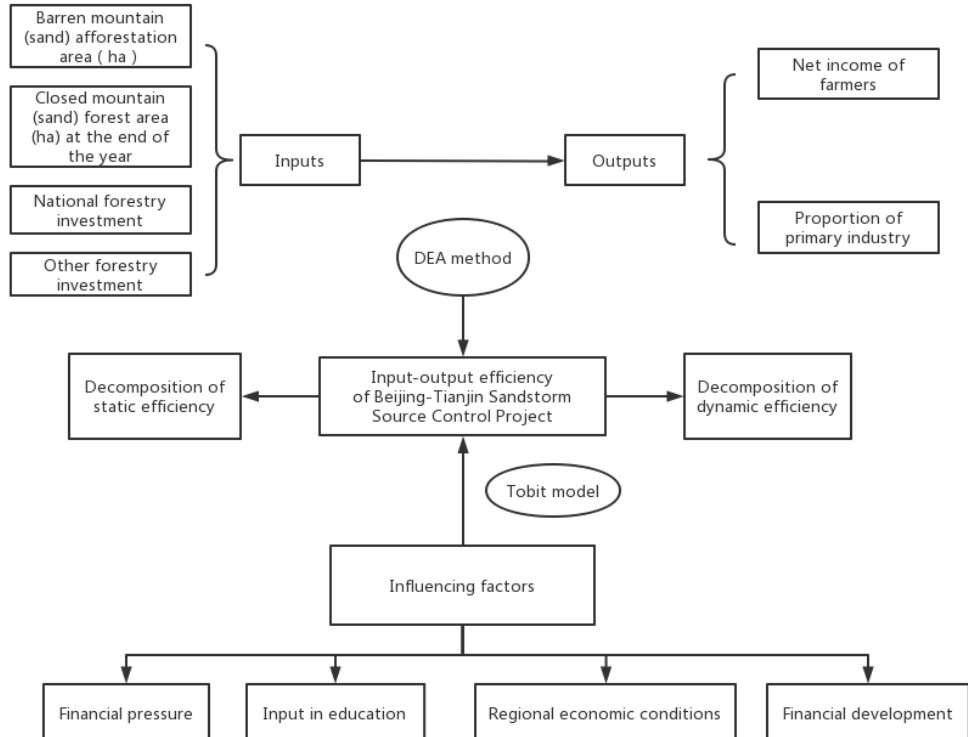

**Figure 1.** The algorithmic structure.

## 2. Research Design

### 2.1. Measurement Method of Input–Output Efficiency

The DEA method is a method to reflect the relative efficiency of each decision-making unit by means of linear programming before determining effective production. Its decision-making units are usually determined by specific province, year, etc. This method judges the effectiveness of decision-making units on the basis of calculating the relative efficiency of decision-making units, and makes comparison among effective decision-making units (A. Charnes et al.) [23]. The most basic model in the DEA method is the CCR model. Assume that the model consists of r sandstorm source control project inputs, s sandstorm source control project outputs, and n sandstorm source control project decision-making units (DMUs). If the engineering input vector of the ith sandstorm source control project decision unit (DMU) is $x_i = (x_{1i}, x_{2i} \ldots \ldots x_{ri})$, and the engineering output vector is $y_i = (y_{1i}, y_{2i} \ldots \ldots y_{si})$, the CCR model can be represented as:

$$s.t \min \left[ \theta - \varepsilon \left( e^T s^- + e^T s^+ \right) \right]; \tag{1}$$

$$s.t \sum_{i=1}^{n} \lambda_i x_i + S^+ = \theta x_0; \tag{2}$$

$$\sum_{i=1}^{n} \lambda_i y_i + s^- = y_0; \tag{3}$$

$$\lambda_j \geq 0, j = 1, 2 \ldots \ldots n; \tag{4}$$

$$s^- \geq 0, s^+ \geq 0. \tag{5}$$

where $x_0$, $y_0$ are the input index and output index of the control engineering of the decision-making unit of any $j_0$th sandstorm source engineering, $\theta$ is the relative efficiency value of the decision-making unit of the sandstorm source control project ($0 \leq \theta \leq 1$), which reflects the effective degree of the whole project, $s^+$ of the governance project is the slack variable, and $s^-$ is the residual variable.

DEA models can be divided into two categories, which are distinguished on the basis of whether the hypothetical conditional returns to scale are variable. This paper adopts the VRS model under the assumption of variable returns to scale. The paper uses the Malmquist index to measure dynamic efficiency, and the expression is as follows:

$$M\left(x^t, y^t, x^{t+1}, y^{t+1}\right) = \left(M^t * M^{t+1}\right)^{\frac{1}{2}} = \frac{D_0^{t+1}\left(x^{t+1}, y^{t+1}\right)_v}{D_0^t(x^t, y^t)_v} * \frac{D_0^{t+1}\left(x^{t+1}, y^{t+1}\right)_c / D_0^{t+1}\left(x^{t+1}, y^{t+1}\right)_v}{D_0^t(x^t, y^t)_c / D_0^t(x^t, y^t)_v} \tag{6}$$

where $D_0^t(x^t, y^t)_c$ represents the distance between $(x^t, y^t)$ and the production possibility set boundary of CRS in phase t and $D_0^t(x^t, y^t)_v$ represents the distance between $(x^t, y^t)$ and the production possibility set boundary of VRS in phase t.

Because DEA is a non-parametric estimation method, it can avoid many of the limitations of parametric methods. However, when used to analyze the input efficiency of the Beijing–Tianjin sandstorm source control project, the DEA method cannot reflect the factors that affect the efficiency. Thus, this paper uses the Tobit panel regression model to conduct in-depth analysis on the factors that affect the efficiency. In this paper, panel data are used to analyze efficiency values. While studying differences between individuals, panel data can improve the degree of freedom and effectiveness between variables, collinearity can be reduced, and deviations can be reduced.

### 2.2. Construction of Regression Model of Influencing Factors

The Tobit model is a model with dependent variables that meet certain constraints.

The maximum likelihood estimation method is used to estimate the Tobit model, and consistent estimates of $\beta$ and $\sigma$ can be obtained (James Tobin, [24]). The Tobit model is used to analyze the influencing factors. The factors affecting the input–output efficiency of the Beijing–Tianjin sandstorm source control project are regarded as independent variables, and the efficiency value calculated by the DEA method is regarded as the dependent variable. As the comprehensive efficiency value calculated by the DEA method is truncated data, it is quite suitable as the restricted dependent variable in the Tobit model.

The regression equation is constructed as follows:

$$C_t^* = \alpha + \sum_{i=1}^{4} \beta_i F_{it} + \varepsilon_t \tag{7}$$

$$\text{When } 0 < C_t \leq 1, \ C_t^* = C_t; \text{ when } C_t \leq 0, \ C_t^* = 0; \text{ and when } C_t > 1, C_t^* = 1 \tag{8}$$

Here, $C_t^*$ is the value of the explanatory variable that is actually entered into Statais and which meets the above conditions, $C_t$ represents the input–output comprehensive efficiency of the Beijing–Tianjin sandstorm source control project in the tth region, F is an explanatory variable representing the factors that affect the engineering efficiency value, $\varepsilon$ represents the random variables, $\alpha$ represents the intercept term, and $\beta$ represents the estimated parameter.

### 2.3. Variable Setting and Data Source

### 2.3.1. Input and Output Indicators for Efficiency Measurement

In the selection of indicators, we refer to western economic theories and research results of forestry input–output efficiency at home and abroad. The French economist

Say believes that the value of any commodity is created by the three production factors of capital, land, and labor. Almost all scholars choose capital as the main input index (Li et al. [25]), while most take natural resources as the input index (Mi et al. [26]). However, the inputs of the Beijing–Tianjin sandstorm source control project engineering mainly include inputs from natural resource endowments and financial resources. This paper uses four indicators, namely, barren mountain (sand) afforestation area (ha), closed mountain (sand) forest area (ha) at the end of the year, national forestry investment (ten thousand RMB), and other forestry investment (ten thousand RMB) to represent the inputs of the Beijing–Tianjin sandstorm source project. In terms of output, in order to reveal the economic benefits of the project, this paper selects the net income of farmers (RMB) and the proportion of primary industry (%) to comprehensively reflect the economic benefits of the output.

2.3.2. Influencing Factor Variables

The influencing variables defined in this paper include fiscal pressure, educational development, regional economic situation, and financial development.

(1)  Fiscal pressure

Fiscal pressure is measured by the proportion of a local government's fiscal expenditure (100 million RMB) to fiscal revenue (100 million RMB). The higher the ratio, the greater the fiscal pressure. The capital of the Beijing–Tianjin sandstorm source control project comes from both central finance and local finance. When the local finance is under great pressure, the fiscal expenditure is large, and the government supports various public undertakings greatly. In addition, Article 38 of the Agricultural Law of the People's Republic of China stipulates that the annual increase in the total fiscal input to agriculture by the central government and local governments at or above the county level shall be higher than the rate of increase in their recurrent fiscal revenues. Therefore, with the increase of fiscal expenditure, more funds will be invested in forestry production to improve forestry production technology. At the same time, the capital of the Beijing–Tianjin sandstorm source control project increases correspondingly, and the applied technology will be more mature. To sum up, it is expected that the fiscal pressure is positively correlated with the efficiency value.

(2)  Input in education

Education investment is measured by the proportion of local education funds (100 million RMB) to the GDP of the region (100 million RMB). The higher the ratio, the more education investment. The greater the investment in education, the higher the average level of knowledge and skills in the region. The greater the knowledge and skills of the specific construction methods and methods of operation included in the Beijing–Tianjin sandstorm source control project, the easier it is to recruit qualified engineering and construction personnel in the region, which is conducive to improving efficiency.

(3)  Regional economic conditions

Regional economic conditions are measured by the proportion of per capita consumption level (RMB) to per capita GDP (RMB). The higher the ratio, the higher the level of economic development. The higher the per capita consumption, the richer the area is, which usually means a better level of economic development. Areas with a high level of economic development are more perfect in the construction of various infrastructure, which is conducive to the construction of sandstorm source control projects and improved output efficiency.

(4)  Financial development

Financial development is measured by the proportion of loan balance at the end of the year (100 million RMB) to the gross regional product (100 million RMB). The higher the ratio, the larger the scale of investment and the higher the level of financial development in the region. The higher the level of financial development, the more efficient the use of

funds, thus making the input and output efficiency of Beijing–Tianjin sandstorm source control project higher and positively contributing to input and output efficiency.

### 2.3.3. Data Sources

Barren mountain (sand) afforestation area (ha), afforestation area of closed mountain and barren land at the end of the year (ha), national forestry investment (ten thousand RMB), and other forestry investment (ten thousand RMB) data were taken from the China Forestry Statistical Yearbook 2003 to 2019. Net income of farmers (RMB), added value of primary industry, fiscal expenditure (one hundred million RMB) and fiscal revenue (one hundred million RMB) of local governments, data on regional education expenditure (100 million RMB), per capita resident consumption (RMB), regional year-end deposit balance (100 million RMB), regional GDP (100 million RMB), and per capita GDP (RMB) were taken from the website of the National Bureau of Statistics. The descriptive statistics of the data are shown in Table 1.

**Table 1.** Descriptive statistics of index data.

| Indicators | Region | The Mean | The Standard Deviation | The Maximum | The Minimum | The Median |
|---|---|---|---|---|---|---|
| Net income of farmers (RMB) | Beijing | 15,700.94118 | 7000.573779 | 28,928 | 5398 | 14,736 |
| | Hebei | 7833.271765 | 3997.299376 | 15,373 | 2685 | 7119.69 |
| | Shanxi | 7107.719412 | 4331.741301 | 16,124.39 | 2299.4 | 5601.4 |
| | Inner Mongolia | 7234.917059 | 4009.272366 | 15,283 | 2132 | 6641.56 |
| Proportion of primary industry (%) | Beijing | 0.008294118 | 0.00339142 | 0.016 | 0.003 | 0.008 |
| | Hebei | 0.119760444 | 0.018852031 | 0.1612 | 0.0921 | 0.1142 |
| | Shanxi | 0.057764706 | 0.007840539 | 0.075 | 0.046 | 0.057 |
| | Inner Mongolia | 0.138529412 | 0.020674274 | 0.178 | 0.108 | 0.138 |
| Barren mountain (sand) afforestation area (ha) | Beijing | 14,010.22 | 6678.82 | 27,399 | 825 | 12,027.5 |
| | Hebei | 11,1583.2 | 69,517.05535 | 326,255 | 38,668 | 99,548 |
| | Shanxi | 51,805.66667 | 41,052.81289 | 165,971 | 4979 | 33,346 |
| | Inner Mongolia | 238,059.8667 | 128,040.8802 | 468,988 | 67,211 | 205,119 |
| Closed mountain (sand) forest area (ha) at the end of the year | Beijing | 87,479.85714 | 35,251.04 | 140,199 | 22,933 | 84,941 |
| | Hebei | 563,179 | 115,985.159 | 731,787 | 310,035 | 565,767.5 |
| | Shanxi | 156,907.2857 | 42,358.34915 | 272,071 | 72,390 | 156,806 |
| | Inner Mongolia | 1,062,554.857 | 454,433.1277 | 1,808,451 | 473,103 | 924,706 |
| National forestry investment (ten thousand RMB) | Beijing | 992,204.8235 | 840,255.8481 | 2,409,673 | 13,269 | 851,235 |
| | Hebei | 475,658.1176 | 301,217.4291 | 1,135,517 | 130,895 | 411,553 |
| | Shanxi | 525,651.2353 | 318,607.402 | 1,036,327 | 164,066 | 378,681 |
| | Inner Mongolia | 954,173.6471 | 479,868.0054 | 1,670,665 | 321,079 | 1,048,171 |
| Other forestry investment (ten thousand RMB) | Beijing | 137,071 | 176,944.4016 | 557,169 | 168 | 46,942 |
| | Hebei | 210,398 | 197,286.9899 | 766,783 | 17,913 | 141,700 |
| | Shanxi | 250,763.8824 | 225,124.8911 | 663,968 | 6205 | 209,063 |
| | Inner Mongolia | 84,429.17647 | 91,151.44861 | 326,942 | 7035 | 48,608 |

As can be seen from the table, the net income of farmers in Beijing is significantly higher than that of the other three provinces and autonomous regions, and the share of primary industry, barren mountain (sand) afforestation area (ha), closed mountain (sand) forest area (ha) at the end of the year, national forestry investment, and other forestry investment in Beijing are significantly lower than that of the other three provinces and autonomous regions. In addition, the barren mountain (sand) afforestation area (ha), closed mountain (sand) forest area (ha) at the end of the year, national forestry investment, and other forestry investment in Inner Mongolia is significantly are higher than the other three provinces and cities.

### 3. Empirical Analysis of Input–Output Efficiency

#### 3.1. Calculation of Static Efficiency

We use four representative provinces (cities, autonomous regions) in the Beijing–Tianjin sandstorm source control project for analysis, respectively, Beijing, Hebei, Shanxi, and Inner Mongolia. When analysing and evaluating the efficiency of decision units, the assumption of variable size is more consistent with the reality, and variable return to scale can become the basic assumption in terms of output indicators. Taking each province as the decision unit (DMU), the DEA model was used to calculate the production efficiency values of each province in each year. Using the DEA method, according to Equations (1)–(5), the efficiency values of Beijing, Hebei, Shanxi, and Inner Mongolia from 2003 to 2019 can be calculated. The comprehensive efficiency values are reflected in the broken line graph shown in Figure 2.

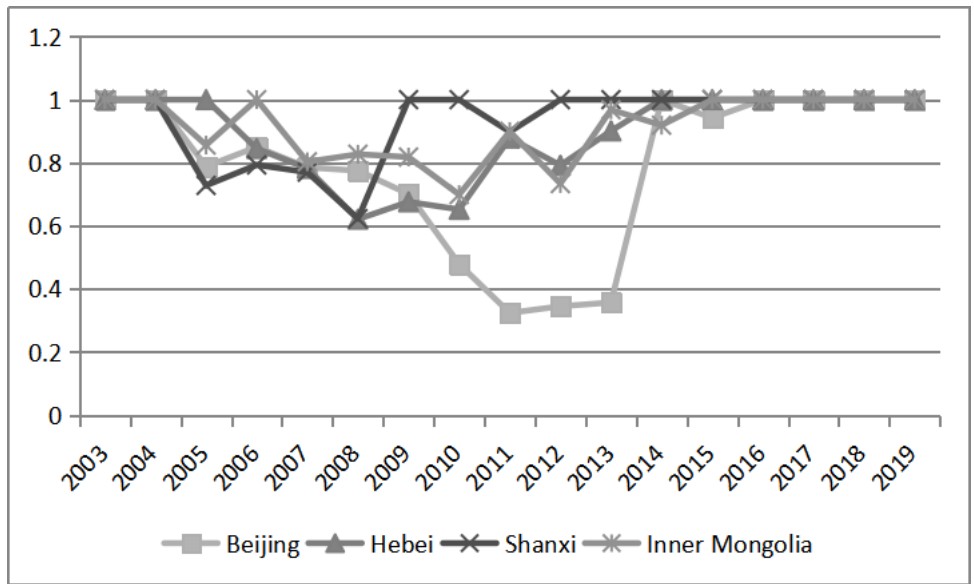

**Figure 2.** Line chart of input–output efficiency of the four provinces from 2003 to 2019.

As can be seen from Figure 2, the input–output efficiency values of the four provinces experienced a u-shaped trend from DEA effective decline to invalid and then back to effective. From the perspective of each province (city and autonomous region), the decline degree of each region is different. Beijing has the largest fluctuation, while Inner Mongolia has a relatively gentle fluctuation.

On the one hand, the u-shaped trend may be determined by the lag of the output benefits of forestry projects. The investment cycle of forestry engineering is long, and it is difficult to see the obvious effect in a short time. The output effect of the resources invested in the past may be observed in the next few years. On the other hand, after project funds are issued, fund managers need to go through a process from inefficient to efficient use of funds, which cannot make the overall efficiency of the project worthy of rapid improvement and needs to be reflected in the follow-up of the project. However, with the progress of the project, there may be a mismatch between the project facilities and the actual situation, and a lag of funds and policies, affecting the efficiency of the project.

The period 2003–2012 was the first phase of the project. It can be seen that before 2012, the four provinces' efficiency values generally decline along with the original DEA effective situation. The reason for this may be that there is a significant increase in input variables. As output indicators, the net income of farmers has steadily increased and the share of the primary sector has steadily decreased. However, input indicators are often related to policy and actual conditions in the year, and can therefore fluctuate widely. For example, as output indicators, the net income of farmers increased from RMB 13,262 in 2010 to RMB 14,736 in

2011, with a growth rate of 11.11%, while the proportion of primary industry decreased from 8.2% in 2010 to 7.8% in 2011, with a decline rate of 4.64%. As input indicators, in 2010, national forestry investment, other forestry investment, barren mountain (sand) afforestation area (ha), and afforestation area of closed mountain and barren land at the end of the year (ha) were RMB 310.665 million, RMB 37.369 million, 7787 hectares, and 106,197 hectares, respectively; in 2011, these indicators were RMB 774.033 million, RMB 179.554 million, 17,753 hectares, and 129,731 hectares, representing a respective increase of 149.15%, 380.49%, 127.98%, and 22.16% over the previous year. In other words, the rate of change of input indicators was much higher than that of output indicators in Beijing in 2011. The situation was similar in other provinces.

The period 2012–2019 was the second phase of the project. After investment increased in 2012, the input–output efficiency value did not increase significantly, although it was constantly reflected in the efficiency value of the following years; thus, it can be inferred that there is a lag in project engineering investment. After 2014, the efficiency of all provinces returned to the effective value. It can be concluded that due to the development of the Phase II regulation project, the resulting financial capital and corresponding improved supporting policies to speed up implementation reversed the situation of insufficient funds and supporting policies with backwards infrastructure, causing the efficiency value in all four provinces to generally rebound.

According to the data in Figure 2, an efficiency value of 1 indicates that the DEA is effective, while a value less than 1 indicates that the DEA is invalid. On the whole, Shanxi Province has the highest comprehensive DEA efficiency of 0.930. Inner Mongolia follows with 0.913, then Hebei with 0.891, while Beijing has the lowest score at 0.785.

From 2003 to 2019, there were four invalid years of comprehensive efficiency in Shanxi Province, which were 2005, 2007, 2008, and 2011. In the years when DEA efficiency is invalid, the efficiency value is generally around 0.8, which is close to the effective frontier, indicating that Shanxi Province's input–output efficiency has always been at a relatively high level. From 2003 to 2019, the efficiency values of Hebei province and Inner Mongolia have similar changes. There were eight ineffective years of DEA efficiency in Hebei Province, concentrated in the period from 2006 to 2013. There were nine years with invalid DEA efficiency in Inner Mongolia, concentrated in the period from 2007 to 2014. When DEA is invalid, the average efficiency value of Hebei province is about 0.77, and that of Inner Mongolia is about 0.84. Generally speaking, the efficiency value of the two provinces is at a relatively high level. On average, the efficiency value of Inner Mongolia is higher than that of Hebei Province.

From 2003 to 2019, there were ten years with invalid DEA efficiency value, respectively, from 2003 to 2013 and 2015. The efficiency values during 2010–2013 were even lower than 0.5, and the average efficiency value of the invalid years was 0.63. In Beijing, the efficiency values in the beginning and ending years were higher, while the efficiency values in the middle development stage were lower.

### 3.2. Calculation of the Change Rate of Economic Efficiency

Input–output efficiency value can reflect the effective degree of a decision-making unit from a static perspective, but it cannot describe the change in input–output index over time. In order to directly reflect the change of comprehensive efficiency during the measurement period, according to Equation (6), Malmquist index is calculated by DEAP software in this paper to reflect the dynamic change rate of economic efficiency. The rate of change of economic efficiency of four provinces (municipalities and autonomous regions) and the whole was calculated from 2004 to 2019, and the results are shown in Table 2.

**Table 2.** Change rate of economic efficiency.

| Malmquist | Total | Beijing | Hebei | Shanxi | Inner Mongolia |
|---|---|---|---|---|---|
| 2003 | - | - | - | - | - |
| 2004 | 1.917 | 10.613 | 1.030 | 1.047 | 1.179 |
| 2005 | 0.615 | 0.309 | 0.915 | 0.738 | 0.685 |
| 2006 | 0.870 | 0.597 | 0.984 | 0.665 | 1.469 |
| 2007 | 0.925 | 1.037 | 1.110 | 0.983 | 0.650 |
| 2008 | 1.070 | 1.907 | 0.720 | 0.662 | 1.442 |
| 2009 | 0.823 | 0.496 | 0.752 | 1.365 | 0.902 |
| 2010 | 0.802 | 1.206 | 1.182 | 0.826 | 0.352 |
| 2011 | 0.712 | 0.595 | 0.802 | 0.870 | 0.620 |
| 2012 | 0.876 | 0.672 | 0.897 | 1.037 | 0.940 |
| 2013 | 0.827 | 0.836 | 0.974 | 1.091 | 0.527 |
| 2014 | 1.090 | 1.684 | 1.087 | 0.925 | 0.838 |
| 2015 | 1.636 | 5.048 | 0.988 | 1.087 | 1.312 |
| 2016 | 0.948 | 0.642 | 0.865 | 1.134 | 1.282 |
| 2017 | 1.061 | 0.649 | 0.957 | 1.094 | 1.866 |
| 2018 | 1.050 | 1.733 | 0.873 | 1.027 | 0.783 |
| 2019 | 1.004 | 0.714 | 1.041 | 1.020 | 1.340 |
| mean | 1.014 | 1.796 | 0.949 | 0.973 | 1.012 |

From 2003 to 2019, the overall mean of the change rate of economic efficiency of the four provinces was 1.014, that is, the average annual increase of efficiency value was 1.4%, and the economic benefit was enhanced. Among them, Beijing's efficiency value increased the most, reaching the level of 79.6%. Efficiency values in Shanxi and Hebei provinces decreased. The efficiency value for Inner Mongolia is 1.20%.

*3.3. Analysis of Decomposition Efficiency Values*

(1)　Static input–output efficiency decomposition analysis

In order to further analyze the reasons leading to the invalidity of DEA in each province, the comprehensive efficiency is decomposed, and the results are shown in Table 3.

**Table 3.** Decomposition table of input–output efficiency.

| Year | Beijing | | | | Hebei | | | | Shanxi | | | | Inner Mongolia | | | |
|---|---|---|---|---|---|---|---|---|---|---|---|---|---|---|---|---|
| | crste | vrste | Scale | rs | crste | vrste | Scale | rs | crste | vrste | Scale | rs | crste | vrste | Scale | rs |
| 2003 | 1.000 | 1.000 | 1.000 | - | 1.000 | 1.000 | 1.000 | - | 1.000 | 1.000 | 1.000 | - | 1.000 | 1.000 | 1.000 | - |
| 2004 | 1.000 | 1.000 | 1.000 | - | 1.000 | 1.000 | 1.000 | - | 1.000 | 1.000 | 1.000 | - | 1.000 | 1.000 | 1.000 | - |
| 2005 | 0.788 | 1.000 | 0.788 | drs | 1.000 | 1.000 | 1.000 | - | 0.728 | 0.788 | 0.924 | drs | 0.856 | 0.861 | 0.995 | irs |
| 2006 | 0.850 | 1.000 | 0.850 | irs | 0.845 | 1.000 | 0.845 | drs | 0.793 | 0.794 | 1.000 | - | 1.000 | 1.000 | 1.000 | - |
| 2007 | 0.785 | 0.789 | 0.995 | irs | 0.782 | 0.972 | 0.805 | drs | 0.770 | 1.000 | 0.770 | irs | 0.803 | 0.824 | 0.974 | irs |
| 2008 | 0.775 | 0.790 | 0.980 | irs | 0.621 | 0.865 | 0.718 | drs | 0.623 | 0.637 | 0.977 | drs | 0.827 | 0.853 | 0.970 | irs |
| 2009 | 0.701 | 0.740 | 0.947 | drs | 0.676 | 0.915 | 0.739 | drs | 1.000 | 1.000 | 1.000 | - | 0.818 | 0.846 | 0.967 | irs |
| 2010 | 0.479 | 0.734 | 0.654 | drs | 0.652 | 0.890 | 0.733 | drs | 1.000 | 1.000 | 1.000 | - | 0.699 | 0.702 | 0.996 | irs |
| 2011 | 0.324 | 0.754 | 0.430 | drs | 0.878 | 0.963 | 0.912 | drs | 0.898 | 0.910 | 0.986 | drs | 0.899 | 1.000 | 0.899 | irs |
| 2012 | 0.345 | 0.818 | 0.421 | drs | 0.792 | 0.933 | 0.849 | drs | 1.000 | 1.000 | 1.000 | - | 0.734 | 0.757 | 0.970 | drs |
| 2013 | 0.357 | 0.909 | 0.392 | drs | 0.901 | 1.000 | 0.901 | drs | 1.000 | 1.000 | 1.000 | - | 0.968 | 0.969 | 0.999 | drs |
| 2014 | 1.000 | 1.000 | 1.000 | - | 1.000 | 1.000 | 1.000 | - | 1.000 | 1.000 | 1.000 | - | 0.919 | 0.931 | 0.988 | drs |
| 2015 | 0.942 | 1.000 | 0.942 | drs | 1.000 | 1.000 | 1.000 | - | 1.000 | 1.000 | 1.000 | - | 1.000 | 1.000 | 1.000 | - |
| 2016 | 1.000 | 1.000 | 1.000 | - | 1.000 | 1.000 | 1.000 | - | 1.000 | 1.000 | 1.000 | - | 1.000 | 1.000 | 1.000 | - |
| 2017 | 1.000 | 1.000 | 1.000 | - | 1.000 | 1.000 | 1.000 | - | 1.000 | 1.000 | 1.000 | - | 1.000 | 1.000 | 1.000 | - |
| 2018 | 1.000 | 1.000 | 1.000 | - | 1.000 | 1.000 | 1.000 | - | 1.000 | 1.000 | 1.000 | - | 1.000 | 1.000 | 1.000 | - |
| 2019 | 1.000 | 1.000 | 1.000 | - | 1.000 | 1.000 | 1.000 | - | 1.000 | 1.000 | 1.000 | - | 1.000 | 1.000 | 1.000 | - |
| mean | 0.785 | 0.914 | 0.847 | | 0.892 | 0.973 | 0.912 | | 0.930 | 0.949 | 0.980 | | 0.913 | 0.926 | 0.986 | |

Note: Comprehensive efficiency (crste) = Technical efficiency (vrste) × scale efficiency (scale); drs represents decreasing scale efficiency and IRS represents increasing scale efficiency.

There are 37 effective DEA comprehensive efficiency values in four provinces (Shanxi 13, Hebei 9, Inner Mongolia 8, Beijing 7). There are a total of 31 invalid DEA comprehensive efficiency values (Shanxi 4, Hebei 8, Inner Mongolia 9, Beijing 10), among which 45.6% of the efficiency values are in an invalid state of technical efficiency and scale efficiency after decomposition, as comprehensive efficiency (CRSTE) = technical efficiency (VRSTE) × scale efficiency (SCALE).

Next, the reasons for the invalidity of the efficiency values are analyzed from the perspectives of technical efficiency and scale efficiency in each province (municipality and autonomous region).

I.    Technical efficiency analysis

For Beijing, the inefficiency of technical efficiency values from 2007 to 2009 is the main reason for the inefficiency of comprehensive efficiency. In 2005, 2007–2010, and 2011–2014 in Beijing and Inner Mongolia Autonomous Region, and in 2005, 2008, and 2011 in Shanxi Province, the reason for the inefficiency of DEA comprehensive efficiency in these years is mainly because both technical efficiency and scale efficiency are invalid, with technical efficiency having a great influence on overall efficiency. A possible reason is that the proportion of state input and other capital input in invalid years in these regions is not coordinated.

II.    Scale efficiency analysis

The inefficiency of the scale efficiency value is the main factor causing the inefficiency of Hebei Province. In Hebei Province from 2007 to 2012 and Beijing from 2010 to 2013, the inefficiency of comprehensive DEA in these regions and years was mainly due to the inefficiency of scale. The main reason is that the scale of investment is unreasonable. For years of scale efficiency increases, increasing investment scale can bring scale efficiency and overall efficiency. For years of decreasing scale, this is due to irrational investment of funds, for which the marginal efficiency of output from excessive investment has reached saturation and cannot continue to increase.

(2)    Dynamic efficiency value decomposition analysis

The change rates of economic efficiency of the four provinces were decomposed, and the results are shown in Table 4.

**Table 4.** Decomposition of productivity factors in the four provinces.

| Year | Beijing | | | | Hebei | | | | Shanxi | | | | Inner Mongolia | | | |
|------|---------|--------|--------|--------|--------|------|------|------|-------|------|------|------|--------|------|------|------|
| | techch [1] | pech [2] | sech [3] | tfpch [4] | techch | pech | sech | tfpch | echch | pech | sech | tfpch | techch | pech | sech | tfpch |
| 2004 | 10.61 | 1.00 | 1.00 | 10.61 | 1.03 | 1.00 | 1.00 | 1.03 | 1.05 | 1.00 | 1.00 | 1.05 | 1.18 | 1.00 | 1.00 | 1.18 |
| 2005 | 0.31 | 1.00 | 1.00 | 0.31 | 0.92 | 1.00 | 1.00 | 0.92 | 0.74 | 1.00 | 1.00 | 0.74 | 0.84 | 1.00 | 0.82 | 0.69 |
| 2006 | 0.60 | 1.00 | 1.00 | 0.60 | 0.98 | 1.00 | 1.00 | 0.98 | 0.67 | 1.00 | 1.00 | 0.67 | 1.20 | 1.00 | 1.23 | 1.47 |
| 2007 | 1.04 | 1.00 | 1.00 | 1.04 | 1.11 | 1.00 | 1.00 | 1.11 | 0.98 | 1.00 | 1.00 | 0.98 | 0.76 | 0.95 | 0.86 | 0.65 |
| 2008 | 1.91 | 1.00 | 1.00 | 1.91 | 0.72 | 1.00 | 1.00 | 0.72 | 0.66 | 1.00 | 1.00 | 0.66 | 1.24 | 1.06 | 1.17 | 1.44 |
| 2009 | 0.50 | 1.00 | 1.00 | 0.50 | 0.75 | 1.00 | 1.00 | 0.75 | 1.37 | 1.00 | 1.00 | 1.37 | 0.90 | 0.84 | 1.00 | 0.90 |
| 2010 | 1.21 | 1.00 | 1.00 | 1.21 | 1.18 | 1.00 | 1.00 | 1.18 | 0.83 | 1.00 | 1.00 | 0.83 | 0.35 | 1.04 | 1.00 | 0.35 |
| 2011 | 0.60 | 1.00 | 1.00 | 0.60 | 0.80 | 1.00 | 1.00 | 0.80 | 0.87 | 1.00 | 1.00 | 0.87 | 0.62 | 1.14 | 1.00 | 0.62 |
| 2012 | 0.67 | 1.00 | 1.00 | 0.67 | 0.90 | 1.00 | 1.00 | 0.90 | 1.04 | 1.00 | 1.00 | 1.04 | 0.94 | 0.77 | 1.00 | 0.94 |
| 2013 | 0.84 | 1.00 | 1.00 | 0.84 | 0.97 | 1.00 | 1.00 | 0.97 | 1.09 | 1.00 | 1.00 | 1.09 | 0.53 | 1.29 | 1.00 | 0.53 |
| 2014 | 1.68 | 1.00 | 1.00 | 1.68 | 1.09 | 1.00 | 1.00 | 1.09 | 0.93 | 1.00 | 1.00 | 0.93 | 0.84 | 0.80 | 1.00 | 0.84 |
| 2015 | 5.05 | 1.00 | 1.00 | 5.05 | 0.99 | 1.00 | 1.00 | 0.99 | 1.09 | 1.00 | 1.00 | 1.09 | 1.31 | 1.00 | 1.00 | 1.31 |
| 2016 | 0.64 | 1.00 | 1.00 | 0.64 | 0.87 | 1.00 | 1.00 | 0.87 | 1.13 | 1.00 | 1.00 | 1.13 | 1.28 | 1.05 | 1.00 | 1.28 |
| 2017 | 0.65 | 1.00 | 1.00 | 0.65 | 0.96 | 1.00 | 1.00 | 0.96 | 1.09 | 1.00 | 1.00 | 1.09 | 1.87 | 1.00 | 1.00 | 1.87 |
| 2018 | 1.73 | 1.00 | 1.00 | 1.73 | 0.87 | 1.00 | 1.00 | 0.87 | 1.03 | 1.00 | 1.00 | 1.03 | 0.78 | 1.00 | 1.00 | 0.78 |
| 2019 | 0.71 | 1.00 | 1.00 | 0.71 | 1.04 | 1.00 | 1.00 | 1.04 | 1.02 | 1.00 | 1.00 | 1.02 | 1.34 | 1.00 | 1.00 | 1.34 |
| mean | 1.80 | 1.00 | 1.00 | 1.80 | 0.95 | 1.00 | 1.02 | 0.95 | 0.97 | 1.00 | 1.00 | 0.97 | 1.00 | 1.00 | 1.00 | 1.01 |

[1] techch—technology progress rate; [2] pech—pure technical efficiency; [3] sech—scale efficiency; [4] tfpch—rate of change in economic efficiency.

It can be seen from the decomposition efficiency value that the technological progress rate is the main factor affecting the change rate of economic efficiency in Beijing, Hebei, Shanxi, and Inner Mongolia. The average technological progress rate of the four provinces (municipalities and autonomous regions) is 1.80, 0.95, 0.97, and 1, respectively, and Beijing is significantly higher than the other three regions. As the technological progress rate

mainly reflects the overall technological improvement of the Beijing–Tianjin sandstorm source control project, the above figures show that the technological improvement and innovation ability of the Beijing Engineering Area is the strongest. This is probably due to the fact that Beijing's level of economic development and the educational and research resources it possesses are far superior to those of the other three regions.

## 4. Empirical Research on Influencing Factors

### 4.1. Description of Variables

As mentioned above, four variables, namely, fiscal pressure, education development status, regional economic development, and financial development, were selected as explanatory variables, and the input–output efficiency values of the Beijing–Tianjin sandstorm source control project in four provinces (municipalities and autonomous regions) were used as explanatory variables for regression. The use of symbols is as follows (Table 5).

**Table 5.** Variable definition and description.

| Variable Types | Variable Symbol | The Variable Name | Variable Declaration |
| --- | --- | --- | --- |
| Explained variable | C | Input-output efficiency | According to the results in Figure 1 of this paper |
| Explanatory variables | $F_1$ | Fiscal pressure | A measure of the ratio of local government expenditure to revenue |
| Explanatory variables | $F_2$ | Educational development status | A measure of education spending as a percentage of GDP |
| Explanatory variables | $F_3$ | Regional economic development | The ratio of household consumption level to per capita GDP |
| Explanatory variables | $F_4$ | Financial development | Proportion of household deposits in regional GDP at the end of the year |

### 4.2. Empirical Analysis

#### 4.2.1. Unit Root Test and Cointegration Test

The panel data of four provinces (municipalities, autonomous regions) from 2004 to 2019 were processed as follows. After the LLC, ADF-Fisher Chi-square, and PP-Fisher Chi-square unit root tests, the results show that the level test of the five series cannot reject the unit root, but after the first-order difference can reject the original hypothesis at the 1% level of significance, so the above series are first-order single integer series. This paper uses the Kao test and Pedroni method for cointegration testing, and the results show that all four variables reject the original hypothesis at a 5% level of significance, which shows that there is a long-term stable relationship between the four influencing factor variables and the explained variables in the four provinces (cities and autonomous regions).

#### 4.2.2. Regression Analysis

The LR test was performed using Stata15 and the results showed that the data should be regressed using a panel Tobit regression with fixed effects. Thus, this paper performs a Tobit regression based on Equations (7) and (8). In order to test the robustness of the regression results, a least squares regression was performed as well, and the results are as follows (Table 6).

**Table 6.** Regression results.

| Variable | Tobit Regression | OLS Regression |
| --- | --- | --- |
| $F_1$: Fiscal pressure | 0.6734984 *** (0.1763233) | 0.1621822 *** (0.0514033) |
| $F_2$: Educational development level | 3.458915 (2.494572) | 0.2814971 (0.1976356) |
| $F_3$: Regional economic development level | 3.280853 ** (1.295633) | 0.5443188 (0.5386912) |
| $F_4$: Regional financial development level | 0.4002577 *** (0.146111) | 0.0871645 (0.0631869) |
| Intercept item $\alpha$ | $-1.805526$ *** (0.6599287) | 0.2915206 (0.175619) |
| $R^2$ | 0.6115 | 0.1550 |

Note: *** represents $p < 0.01$, ** represents $p < 0.05$; The number above the brackets is the coefficient, and the number inside the brackets is the standard deviation.

According to the regression results, the parameter signs of the two regression methods are consistent, therefore, the results are robust. Under Tobit regression, fiscal pressure and regional financial development level are significant at a 1% confidence level, while regional economic development level is significant at a 5% confidence level. Among them, the level of education development has the greatest impact on efficiency, followed by the level of regional economic development. The influence of fiscal pressure and regional financial development level is small.

The level of education development has a strong positive impact on efficiency. On the one hand, the Beijing–Tianjin sandstorm source control project is a comprehensive environmental control project. In addition to various afforestation projects, grassland management, and small integrated watershed management projects, it needs many supporting economic facilities. The complexity of engineering determines its strong demand for theoretical and technical talent in different fields. However, regions with developed education have a sufficient talent supply, and it is easy to hire qualified talent to carry out engineering construction, saving recruitment costs and thus improving capital efficiency. On the other hand, a higher the education development level accompanies stronger awareness of environmental protection.

At the same time, it is easier to accept and master the use of modern science and technology for farming and cultivation. As a result, there is relatively little resistance to the implementation of grazing and reforestation projects. It is faster in achieving a change in production and lifestyle, resulting in faster poverty eradication and improved economic benefits of the project. Regional economic development level has a strong positive impact on efficiency. This may be because more economically developed areas have better infrastructure development, and it is easier to carry out construction of projects, which has a positive effect on efficiency value.

Fiscal pressure has a positive effect on efficiency, although the effect is not obvious. This may be because greater fiscal pressure means higher fiscal expenditure, that is, the government has invested heavily in various public services and engineering construction. However, due to the limited fiscal resources allocated to the Beijing–Tianjin sandstorm source control project, although it can promote the improvement of comprehensive efficiency, it cannot bring more obvious effects. For the level of financial development, the more developed the finance, the better the various investment and financing channels, which can enhance the efficiency of capital use.

## 5. Policy Recommendations

Based on the above conclusions, this paper puts forward the following policy suggestions:

(1) Persisting in the control of sandstorm sources, and actively exploring different technologies and management models.

Because the output effect of the sandstorm source control project has a lag, the relevant departments no longer promote it because they cannot achieve obvious results in a short period of time. At the same time, due to the technical progress rate mainly affecting the economic efficiency rate of change in the project, it is unclear when to carry out the engineering technology and management mode of optimization and innovation. The project should adjust measures to local conditions. A variety of models should be used to carry out the project, such as the combination of forestry, water, and agriculture management models, estate economic management model, etc. By organically combining sand control, water conservancy construction, and agricultural cultivation, the ecological and economic benefits are unified.

(2)   Rationally adjust the structure of investment and the amount of input, and establish an input mechanism with dynamic changes.

At present, the average comprehensive efficiency of the four regions from 2003 to 2019 has not reached DEA efficiency because of an unreasonable investment structure and too much or too little input. In terms of investment structure, the management of funds involves many departments, such as forestry, water conservancy, and agriculture, while these departments lack of unified guidance and effective coordination, causing the low service efficiency of these funds. It is suggested that the funds be allocated on the basis of a reasonably determined budget, and that the corresponding standards be determined according to the different costs of treatment in each region, in order to maximize the efficiency of the funds. At the same time, because the resource endowments and socio-economic conditions of the project area are constantly changing, it is suggested to regularly evaluate and determine the subsidy standard and establish a dynamic investment mechanism.

(3)   Increase investment in relevant education, and pay equal attention to theoretical and technical personnel.

The improvement of education level has a great impact on the improvement of efficiency. It is necessary to increase the investment of education funds, especially to provide special funding support to local forestry colleges. In terms of input in education to support the training of human resources, attention should be paid to both the cultivation of high-end theoretical talents and to the cultivation of technical professionals with a solid foundation in order to promote technical research and exploration of afforestation in difficult areas as well as the improvement of project construction quality.

## 6. Conclusions

This study first uses the DEA method to calculate the input–output efficiency of Beijing, Hebei, Shanxi, and Inner Mongolia from the perspective of economic benefits. The results show that the regional efficiency values all experienced a u-shaped trend, first declining and then rising, showing the lagging characteristics of the output of Beijing–Tianjin sandstorm source projects. From 2003 to 2019, the average comprehensive efficiency values of the four places were all less than 1, which did not reach DEA effectiveness. The invalidity of the efficiency values is caused by both technical efficiency and scale efficiency, which indicates the existence of an unreasonable capital structure and unreasonable capital input. The change rate of economic efficiency was calculated by Malmquist index, and the result showed that the mean change rate of economic efficiency of the four provinces (municipalities and autonomous regions) was 1.014 during the measurement period, that is, the average annual increase of efficiency was 1.4%. The decomposition of the rate of change in economic efficiency found that the rate of technological progress is the main influencing factor. The highest rate of change in average economic efficiency in Beijing is explained by the fact that it has the highest rate of technological progress.

In terms of influencing factors, through Tobit regression it was found that education development level and economic development level have a significant positive effect on input–output efficiency, fiscal pressure may promote efficiency values with a smaller effect, and financial development has a dampening effect on efficiency.

**Author Contributions:** Conceptualization, X.G.; methodology, X.G.; software, Y.C.; validation, Z.L. and J.Y.; formal analysis, Y.C.; investigation, Z.L. and J.Y.; resources, X.G.; data curation, Y.C.; writing—original draft preparation, Z.L. and J.Y.; writing—review and editing, Y.C.; visualization, Y.C.; supervision, X.G.; project administration, X.G. All authors have read and agreed to the published version of the manuscript.

**Funding:** This research was funded by projects of the National Natural Science Foundation of China (No. 71703007) and Chinese National Funding of Social Sciences (No. 19BGL052).

**Institutional Review Board Statement:** Our manuscript does not contain data derived from a questionnaire/interview/survey/experiment involving human participants or data/animal subjects.

**Informed Consent Statement:** Informed consent was obtained from all subjects involved in the study.

**Data Availability Statement:** Data can be provided upon request from the corresponding author.

**Conflicts of Interest:** The authors declare no conflict of interest.

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
