# Peer review of "Input–Output Efficiency of the Beijing–Tianjin Sandstorm Source Control Project and Influencing Factors"

_sustainability, doi:10.3390/su14148266_

Round 1
Reviewer 1 Report
Although this study has the importance and interesting points of the topic, there is a lot of room for improvement in the selection of the following variables and the validity of the model.
- The rationale and explanation for the relationship between fiscal pressure and efficiency are weak and unclear. The rationale and explanation for including this variable should be presented.
- Efficiency value shows U shape, and it is interpreted because of lagging characteristics of forestry industry. It can be understood by the explanation that the efficiency value increases after a certain period of time, but it does not explain the reason why it should be lower than before in the beginning. A more convincing explanation for this should be provided.
- The deposit ratio is chosen for financial development, and this variable is negatively correlated with consumption level (regional economic condition). Therefore, there are problems in the selection of variables and the validity of the model.
- Natural endowment variables are selected as inputs, and those variables are factors that we cannot control, so it is more suitable for environment variables.
Reviewer 2 Report
The paper analyses the effectiveness of the Beijing-Tianjin Sandstorm Source Control Project from economic point of view. The authors refer to lots of paper dealing with it from ecological point of view.
Seeking in Google Scholar for Beijing-Tianjin Sandstorm Source Control Project” 140 matches were found. Among them several are in Chinese. As the project costed a huge amount of money, presumably its economic efficiency has already been analysed.
The paper uses DEA (Data Envelopment Analysis), which is a generally accepted method.
The input variables are Barren mountain afforestation, closed mountain forest area, national forestry investment, other forestry investment, the output variables are incomes of farmers, share of primary industry. I think that it would have been useful to involve in the investigation other characteristics, too (for example the value of the produced agrarian products, the number of population), supposing data are available. It is interesting for me that the author performed a truncated linear regression between the efficiency and some other variables (education, financial development, …) for a deep analysis, it would be worth increasing the list of variables.
Questions:
Why were these factors chosen by the authors for DEA?
Fig 1 shows that the efficiency is decreasing for some time, then in the second phase it reaches 1.
Can you explain the reason of the extremely large decrease in Beijing region?
The paper contains lots of mistypes, for example:
In the title: Contorl is written, the name of the 4rd author, line 12 Four instead of Four, Inner Mongolia is mentioned in four different forms, line 136 capital S and small s . It is disturbing that in line 136 y contains the output variables, and in line 176 y is the efficiency (please unify the notations)
Table 3 contains mistypes: for example, Beijing 2006 rs is not decreasing, and so on.
Summarizing: the paper is a medium-level economic analysis. The results are of local interest. I suggest its acceptation after minor revision.
Reviewer 3 Report
The topic is interesting as a real life based problem study, but need to improve in the following issues:
(1) There is several formulae like pp. 134-140, 150-154, 175-177 etc. The formulae should be numbered. Where they com, cite the necessary references, and also mention how they are used.
(2) In table 3 we see some data is input by authors. What is the data source ? How the data is collected, cite properly.
(3) The paper seems to full of table. Rather than table give some figure and explain the figure briefly.
(4) Policy recommendation should be a separate section before the conclusion section,
(5) Structure of the paper should be added in introduction section for readers interest.
(6) How the total problem is solved may be shown by a algorithmic structure. It should be added.
(7) Literature review is not done in proper way. Here few more importance related paper should be cited and make a comparison with your work.
Round 2
Reviewer 1 Report
Although the author presented some convincing explanations and grounds for the last point, there are still rooms for improvement in terms of the selection of the following variables and the validity of the model, and these problems require major improvement efforts.
- The result according to financial pressure results in forest investment, and this value is already used as an input variable for efficiency calculation. Therefore, it is considered inappropriate to use this value as a variable that affects the efficiency again.
- Efficiency value shows a U shape, and it is interpreted as the lagging characteristics of the forestry industry. The author explains that the initial efficiency value is gradually lower than before because of the marginal decreasing effect. However, marginal decreasing effect occurs after some optimal level of capacity utilization. That is, the addition of any larger amounts of a factor of production will inevitably yield decreased per-unit incremental returns. The What this means is that all DMUs are already in an optimal state of economic output at the beginning of the project. Therefore, the author's explanation lacks validity.
- The deposit rate is chosen for financial development, and it is said that this variable does not mean only saving, but at the end of page 16, it is defined as meaning the bank balance. In addition, a small bank balance is interpreted as consumption on investment assets, but it is unreasonable to interpret it this way. Also, looking at the results shown in the analysis, this value is negatively correlated with consumption. Therefore, there are problems in the selection of variables and the validity of the model.
Reviewer 3 Report
The paper is now acceptable in its current form.
Author Response
Thanks for your suggestions. We have made some modifications to the English expressions.
Round 3
Reviewer 1 Report
The authors refined the manuscript accordingly.